# *Slc7a11* Modulated by *POU2F1* is Involved in Pigmentation in Rabbit

**DOI:** 10.3390/ijms20102493

**Published:** 2019-05-20

**Authors:** Yang Chen, Shuaishuai Hu, Lin Mu, Bohao Zhao, Manman Wang, Naisu Yang, Guolian Bao, Cigen Zhu, Xinsheng Wu

**Affiliations:** 1College of Animal Science and Technology, Yangzhou University, Yangzhou 225009, China; yangc@yzu.edu.cn (Y.C.); 18852726848@163.com (S.H.); mulin912@126.com (L.M.); zhao598841633@163.com (B.Z.); wmm171717@126.com (M.W.); yangnaisu@foxmail.com (N.Y.); 2Joint International Research Laboratory of Agriculture & Agri-Product Safety, Yangzhou University, Yangzhou 225009, China; 3Animal Husbandry and Veterinary Research Institute, Zhejiang Academy of Agricultural Sciences, Hangzhou 310000, China; GuolianB@126.com; 4Jinling Rabbit Farm, Nanjing 210005, China; js_zhucigen@foxmail.com

**Keywords:** Rex Rabbit, melanocyte, pigmentation, *Slc7a11*, *POU2F1*

## Abstract

Solute carrier family 7 member 11 (*Slc7a11*) is a cystine/glutamate xCT transporter that controls the production of pheomelanin pigment to change fur and skin color in animals. Previous studies have found that skin expression levels of *Slc7a11* varied significantly with fur color in Rex rabbits. However, the molecular regulation mechanism of *Slc7a11* in pigmentation is unknown. Here, rabbit melanocytes were first isolated and identified. The distribution and expression pattern of *Slc7a11* was confirmed in skin from rabbits with different fur colors. *Slc7a11* affected the expression of pigmentation related genes and thus affected melanogenesis. Meanwhile, *Slc7a11* decreased melanocyte apoptosis, but inhibition of *Slc7a11* enhanced apoptosis. Furthermore, the *POU2F1* protein was found to bind to the −713 to −703 bp region of *Slc7a11* promoter to inhibit its activity in a dual-luciferase reporter and site-directed mutagenesis assay. This study reveals the function of the *Slc7a11* in melanogenesis and provides in-depth analysis of the mechanism of fur pigmentation.

## 1. Introduction

The fur color of mammals mainly depends on melanin deposition, and melanogenesis is mainly regulated by melanocytes. The production of different types of melanin by melanocytes, together with different distribution of these pigments, result in a variety of hair colors in mammals [1]. Related genes, such as *TYR*, *TYRP1*, *ASIP*, *MITF*, and *CREB1*, have been found to regulate melanin deposition [2,3]. Previously, by using the transcriptome sequencing (RNA-Seq), a significant difference was found in the expression of the *Slc7a11* gene in the skin of rabbits with different fur colors. It was speculated to be involved in fur pigmentation [4].

In the melanogenesis pathway, both eumelanin and pheomelanin are derived from a common precursor named dopaquinone [5]. Cystine or glutathione is required for the production of pheomelanin and it is xCT, the protein encoded by the *Slc7a11* gene, that acts as a vector to transport extracellular cystine into the cell and maintain normal intracellular glutathione levels. Pheomelanin and eumelanin together form a mixed pigment that determines the skin and fur color of animals [6,7,8]. In the hair of *Slc7a11* gene-mutated mice (sut), the level of pheomelanin was significantly decreased, while the eumelanin level was substantially unchanged, so that the wild-type mice with yellow background appeared gray [9]. The sut mutation results in a huge deletion in the *Slc7a11* gene, but similar deletions could not be found in this region of Rex Rabbits with six different fur colors, including black (BL), chinchilla (CH), white (WH), brown (BR), protein yellow (PY), and protein chinchilla (PC). SNPs in the exon region of *Slc7a11* were also scanned, but no mutation site was found. This indicates that *Slc7a11* is highly conserved in the population (data not shown). Currently, studies on the functions of *Slc7a11* mainly focus on its important roles in cell proliferation [10], oxidative stress response [11], and Alzheimer’s disease target treatment [12]. Research studying its regulatory mechanisms is focused on microRNAs affecting cancer development and apoptosis by targeting and regulating *Slc7a11* [10,13]. Few studies regarding melanin deposition have been reported.

To explore the molecular regulation mechanism of *Slc7a11* in the melanin deposition of Rex rabbit fur, rabbit melanocytes were isolated and identified. The expression pattern of *Slc7a11* in Rex rabbits with different fur colors was analyzed. Furthermore, we verified that *POU2F1* has an important regulatory role in the transcriptional activation of the *Slc7a11* gene promoter. This result provides a theoretical basis for further analysis of the deposition mechanism of the fur pigmentation as well as for the transformation of fur color in animals.

## 2. Results

### 2.1. Separation and Identification of Rabbit Melanocytes

The back skin of black Rex rabbits was collected and cells separated by a two-step enzyme digestion method. After 12 h of isolation, the keratinocytes were found to have a cobblestone-like appearance and accounted for the majority of cells. The melanocytes, which had the unique bi-polar dendritic morphology, were small in number. However, as cells grew, the keratinocytes gradually died out, the melanocytes continued to divide, and the cell culture became consistent in appearance. When the cells were passed to the third generation, the keratinocytes were almost absent. The melanocytes were dominant with special growing follicles and strong refraction (Figure 1a).

The melanocyte marker genes *MITF*, *TYR*, and *TYRP1* were detected by semi-quantitative PCR (Figure 1b). The isolated cells stained with L-DOPA staining contained brown or black particles (Figure 1c). Immunocytochemical staining of S-100, TYR and TYRP1 revealed that these markers were expressed in melanocytes. Compared with the negative control, S-100 staining showed the cytoplasm and dendrites were positively stained brown. The nucleus was brownish yellow in the TYR staining, and light brown in the TYRP1 staining (Figure 1d). This indicated that the rabbit melanocytes were successfully isolated and identified, providing experimental materials for this study.

### 2.2. Analysis of Slc7a11 Gene Expression in Rex Rabbit Skin with Different Fur Colors

The *Slc7a11* cDNA sequence, including 31 bp 5′UTR, 1509 bp open reading frame (ORF), and 132 bp 3′UTRs (poly-[A] tail included), was obtained using RACE and cloning techniques and submitted to GenBank (Accession number KY971639.1) (Figure 2a). The localization of the Slc7a11 protein in the skin tissue of Rex Rabbits was determined by immunohistochemistry. Blue positive reactions were detected in the epidermis, hair bulbs, and hair root-sheaths, with different shades of color, suggesting *Slc7a11* was widely expressed (Figure 2b). It was found that the expression level of the *Slc7a11* gene was highest in skin with PY color, which was 3.7 times that seen in WH. The differences between PY and WH, as well as between PC and WH, were significant (*p* < 0.01) (Figure 2c). Wes system analysis showed that the Slc7a11 protein was expressed in all skin tissues. The protein expression level was the highest in PY skin, and the lowest in the WH (Figure 2d,e). The mRNA and protein expression levels of *Slc7a11* in Rex rabbit skin with different fur colors showed a significant positive correlation (R = 0.874, *p* < 0.05).

### 2.3. Effect of Slc7a11 Gene Expression on Melanin Deposition

In order to further analyze the mechanism of the *Slc7a11* gene in melanogenesis, *Slc7a11* siRNA interference and overexpression were performed in melanocytes, and qRT-PCR and Wes were used to detect mRNA and protein expression levels of pigment-related genes such as MITF and TYR. The results showed that siRNA-2 and siRNA-3 interferences were significantly lower than that of the blank group (*p* < 0.05), and siRNA-3 had the best effect (Figure 3a,b). pEGFP-N1-Slc7a11 was expressed in melanocytes and the expression of *Slc7a11* was significantly increased in these cells (*p* < 0.01) (Figure 4a,b).

When *Slc7a11* was overexpressed or inhibited, the mRNA (Figure 3c and Figure 4c) and protein expression levels (Figure 3d and Figure 4d) of genes involved in the melanogenesis pathways (such as *MITF*, *TYR*, *TYRP1*, *CREB1*, and *ASIP*) also changed significantly. There was a significant positive correlation between mRNA and protein expression (*p* < 0.05), which was consistent with the changes in the expression of *Slc7a11*. Compared with the control group, the melanin level increased when *Slc7a11* was overexpressed; whereas when *Slc7a11* was inhibited melanin level decreased (Figure 3e and Figure 4e). The results suggested that *Slc7a11* affects the expression of pigmentation-related genes such as *TYR* and *MITF*, and thus affects melanogenesis by melanocytes. We examined the apoptosis rate in melanocytes after transfecting these cells with siRNA-Slc7a11 and pcDNA3.1-Slc7a11 and found that *Slc7a11* decreased melanocyte apoptosis, but inhibition of *Slc7a11* enhanced apoptosis (Figure 3f and Figure 4f).

### 2.4. Identification of the Core Region of the Slc7a11 Promoter and Key Transcription Factor POU2F1

In order to further reveal the regulatory mechanism of the *Slc7a11* gene, the promoter sequence 2499 bp before the start codon of *Slc7a11* was cloned. Firstly, by predictive analysis of potential transcription factors in the *Slc7a11* promoter region, four deletion vectors (P1–P4) were constructed. Dual-luciferase assays showed that the activities of P2 and P3 were comparable (*p* > 0.05). P2 activity was significantly lower than that of P1, and the activity of P4 was significantly lower than that of P3 (*p* < 0.05), indicating that the deletion of −969~−469 bp and −2469~−1969 bp decreased the activity significantly. The results suggested that there were two active regions in the *Slc7a11* promoter, −969~−469 bp and −2469~−1969 bp, respectively (Figure 5a).

In order to further identify the core transcription factor binding region, a series of deletion vectors were constructed targeting the −2469~−1969 bp and −969~−469 bp fragments: P8, P9, P10 for the −2469~−1969 bp region, and P5, P6, and P7 for the −969~−469 bp region. The activities of P8, P9, and P10 were similar by luciferase assay (*p* > 0.05) (Figure 5b). However, the activities of P5 and P6 were comparable but significantly lower than that of P7, indicating that the activity was significantly reduced with the deletion of the −769 to −619 bp region (*p* < 0.05) (Figure 5c), suggesting that −769~−619 bp is the core transcription region of *Slc7a11*. The predicted *POU2F1* binding site (−713 to −703 bp) was found in the −769 to −619 bp region (Figure 5d). It was found that the promoter activity of *Slc7a11* was significantly increased after site-directed mutagenesis (*p* < 0.01), indicating that *POU2F1* inhibited the promoter activity of *Slc7a11* (Figure 5e).

To further determine whether *POU2F1* binds to this site of the *Slc7a11* promoter, an EMSA experiment was performed using the nuclear protein of melanocytes (Figure 5f). The 3rd lane showed that the biotin-labeled probe for *POU2F1* could bind to the nuclear protein to form a complex band. No band in the 4th lane suggested that mutated *POU2F1* could not bind to a nuclear protein to form a complex. The results together revealed that *POU2F1* could bind to the *Slc7a11* core promoter region, and a competitive EMSA experiment was performed to further determine whether the binding was specific (Figure 5g). A complex band in Lane 1 indicated that the probe was able to bind to the nuclear protein. The 2nd and 3rd lanes were cold-competitive reactions with unlabeled normal probes. No bands were observed, indicating that the unlabeled normal probes were competitively bound to the nuclear protein due to their large amounts, meaning the biotin-labeled probe could hardly bind to the protein. The 4th and 5th lanes were cold-competitive groups with unlabeled mutant probes. The unlabeled normal probe did not bind to the protein after mutation, and thus did not compete with the biotin-labeled normal probe, producing protein-probe complex bands. The results confirmed that the POU2F1 protein could bind to the −713 to −703 bp region and inhibit the activity of the *Slc7a11* promoter.

## 3. Discussion

Skin melanocytes are located on the basement membrane between the epidermis and the dermis. Skin melanocytes have been isolated from humans, pigs, and other animals using 0.25% trypsin and successfully cultured [14,15]. In a preliminary study, the back skin of rabbits was digested with trypsin, but the results were not satisfactory, indicating that the use of trypsin alone did not result in sufficient digestion or separation of melanocytes from the skin of hairy animals. However, the use of Dispase II enzyme digestion, allowed us to obtain pure melanocytes for the first time, which laid the foundation for our later work.

Fur color is controlled by different genes in the process of pigment biosynthesis. The differences in color are mainly due to the different ratios between pheomelanin (red and yellow) and eumelanin (black) [16,17,18]. The main model currently proposed is that the ratio of eumelanin/pheomelanin in mammalian pigments is solely or indirectly regulated by the activity of tyrosinase—the rate-limiting enzyme of melanin synthesis [19]. This model inferred that in the presence of low concentrations of tyrosinase, dopaquinone reacts with cysteine to produce cysteine dopa, thereby increasing the level of pheomelanin [20]. However, this pattern is not yet fully understood.

Studies have confirmed that the xCT transporter encoded by the *Slc7a11* gene is crucial for the regulation of pigments and can directly affect the increase of pheomelanin [9]. Based on previous studies, Rex Rabbits with a variety of natural fur colors were used to explore the expression pattern of *Slc7a11* gene in dorsal skin tissues with different fur colors. xCT was expressed in the epidermis, hair bulb, and hair root-sheaths of the skin tissues examined by immunohistochemistry. It is known that melanocytes in the skin are often distributed in different regions when matured and that the only place where melanin can be supplied to the hair shaft is the hair bulb [21,22,23]. In this study, it was found that the expression sites of the *Slc7a11* gene were consistent with the distribution of melanin, suggesting that the protein was related to the formation of Rex Rabbit fur color. Moreover, the *Slc7a11* gene had the highest expression level in protein yellow coat, and the lowest level in white coat, by real-time quantitative and Wes system analyses. It is speculated that *Slc7a11* affects the formation of cystine, which is reduced to cysteine, and thus alters the production of pheomelanin. This is consistent with studies on alpaca and sheep [24,25].

Knockdown and overexpression analyses of *Slc7a11* confirmed that this gene can affect the expression of *TYR*, *MITF*, *TYRP1*, and *ASIP* in the melanogenesis pathway [3,26]. *TYR* is a key enzyme in melanin formation and its expression level and activity determine the rate and yield of melanin production. Upon activation, *TYR* catalyzes the hydroxylation of tyrosine to L-3,4-phenylalanine (DOPA), which is rapidly oxidized to form dopaquinone [27]. *MITF* regulates the expression of the tyrosine gene family and participates in melanin production [28]. Studies have shown that the expression of *TYRP1*, *ASIP* and *CREB1* determines skin melanin deposition [29,30,31]. The *Slc7a11* gene affects the transcription of genes involved in melanogenesis, which is closely related to the formation of fur color. Moreover, *Slc7a11* decreases melanocyte apoptosis and further affects melanogenesis of melanocytes. These results confirmed that *Slc7a11* is closely related to the formation of Rex rabbit fur color. The regulatory factors required for such expression patterns would be the next research objective.

Chintala et al. found that the mouse light gray (sut) mutation was due to the inhibition of phaeomelanin production. It was caused by a large deletion of the *Slc7a11* gene starting from the 11th intron crossing the 12th exon into the region adjacent to the Pcdh18 gene. This resulted in a change in the 3′ end of *Slc7a11* transcription [9]. Based on these results, the identification of similar deletions in the natural populations of Rex Rabbits with six fur colors was carried out. Unfortunately, no such large fragment deletions were found in similar areas. Exon scanning did not show any SNPs sites, indicating that *Slc7a11* is relatively conserved in the population. Similar results have been seen in humans and sheep [25,32,33]. The regulatory mechanism of *Slc7a11* is unknown.

Furthermore, the promoter series deletion vector dual-luciferase was used to search for the −769 to −619 bp transcription core region of *Slc7a11*. We also confirmed that POU2F1 protein binds to the −713 to −703 bp region of the *Slc7a11* promoter to inhibit its activity. *POU2F1*, also known as Octamer Transcription factor-1 (Oct-1), is a widely expressed POU protein factor. Recent studies suggest that it can regulate target genes associated with processes such as oxidation, anti-cytotoxicity, stem cell function, and cancer development, etc. [34,35]. Ethanol has been reported to increase the expression of *Slc7a11* by reducing the binding of *POU2F1* to the *Slc7a11* gene promoter [36]. In this study, *POU2F1* was found to specifically bind to the *Slc7a11* promoter and inhibit its transcription. Together with the previous finding that *Slc7a11* promotes melanin cytochrome deposition, *POU2F1* can be used as a target for artificial modification of animal fur colors.

## 4. Materials and Methods

### 4.1. Primary Separation and Culture of Rabbit Melanocytes

Rabbits were injected with anesthetic on the back, and a piece of the back skin (1.5 cm × 1.5 cm) was dissected. Any particles on the skin surface and subcutaneous connective tissue were removed. After tissue collection, the wounds were treated with iodine tincture. After the experiment, rabbits were anesthetized by an intraperitoneal injection of sodium pentobarbital (50 mg/kg).

The skin sample was digested with 0.25% DispaseII enzyme digestion solution (Sigma, Darmstadt, Germany) for 14–16 h at 4 °C. The epidermis was gently peeled off from the dermis, cut into small pieces, and digested with 0.25% trypsin (Gibco, Carlsbad, CA, USA) for 8 min at 37 °C. The sample was then filtered through a 200 mesh filter and the supernatant discarded. The cells were resuspended in M254 medium (Gibco, Carlsbad, CA, USA) and incubated at 37 °C in a 5% CO_2_ incubator. The cells were digested with 1 mL of 0.25% trypsin and subcultured.

This study was carried out in accordance with the recommendations of Animal Care and Use Committee at Yangzhou University. The protocol was approved by the Animal Care and Use Committee at Yangzhou University (Yangzhou, China, 24 October 2017, No. 201710001).

### 4.2. DOPA Staining

Melanocytes in the logarithmic growth phase were used to prepare sterile cell culture slides. The inoculated 24-well plates were cultured for 3 days and treated with 1 mL of 4% paraformaldehyde fixative (Solarbio, Beijing, Tongzhou, China) for 30 min at 4 °C. The plates were washed 3 times with pre-cooled PBS prior to the addition of L-DOPA (Sigma). After incubation in L-DOPA for 1 day, the incubation solution was renewed and the plates were further incubated at 37 °C for 12 h, with constant observation once every 30 min. The plates were washed with PBS once the staining was complete and observed under a microscope [37].

### 4.3. Immunostaining

Melanocytes in the logarithmic growth phase or skin tissues from Rex rabbits of 6 different fur colors were used to prepare slides. The slides were incubated with primary antibodies Slc7a11 (1:500 rabbit polyclonal, Abcam, Cambridge, UK), S-100 (1:500 mouse monoclonal, Boster, Wuhan, China), TYRP1 (1:250 rabbit polyclonal, Abcam), TYR (1:1000 rabbit polyclonal, Abcam) overnight at 4 °C, with PBS as a negative control. The slides were subsequently incubated with IgG secondary antibody (1:2500 goat polyclonal, Abcam) at 37 °C for 20 min and developed for 3–5 min at room temperature in the dark with freshly prepared DAB solution (Boster). The slides were observed under a microscope.

### 4.4. RACE and Cloning of Slc7a11 Gene

Three specific 5′ RACE primers and two 3′ RACE primers were designed according to the Race kit instructions (Invitrogen & Clontech, Carlsbad, CA, USA) (Table 1). The full-length cDNA sequence of the *Slc7a11* gene was assembled based on known sequences and 5′ and 3′ RACE results, and submitted to NCBI (Accession no.: KY971639.1). The *Slc7a11* cDNA was reconstructed into the pEGFP-N1 vector with restriction enzymes HindIII and SacII.

### 4.5. Knockdown of Slc7a11 by siRNA

Fluorescently labeled siRNAs (with 5′ FAM modification) and Negative Control siRNAs were purchased from Shanghai GenePharma Co., Ltd (Table 2). When melanocyte confluence reached about 65%, the siRNA oligo/Lipofectamine™ 2000 (Invitrogen, Carlsbad, CA, USA) complex at a ratio of 1:2 was prepared for transfection. After 24 h, the transfection efficiency was examined by fluorescence microscopy.

### 4.6. Real-Time PCR

Real-time PCR was carried out using SYBR^®^ Premix Ex Taq™ II (TaKaRa, Dalian, China) on an Applied Bio systems 7500 Real-Time PCR System with the following program: 1 cycle at 95 °C for 30 s, followed by 40 cycles of 95 °C for 5 s, and 60 °C for 30 s. The primers used for detection of gene expression are shown in Table 3. Each sample was measured three times, and the results were normalized to GAPDH. The relative expression of the target gene was calculated by the ΔΔCt method; namely, the fold difference between the target gene and the reference gene (experimental group)/the fold difference between the target gene and the reference gene (control group) $=2^{{(}_{\Delta }{\mathrm{Ct}}^{\mathrm{experimental}}{-}_{\Delta }{\mathrm{Ct}}^{\mathrm{control}})}=2^{{-}_{\Delta \Delta }\mathrm{Ct}}$.

### 4.7. Simple Western Analysis

Pre-cooled RIPA lysis buffer (Sigma) was mixed with PMSF (with a final concentration of 1 mM) and added to the tissue or cell samples, which were centrifuged at 10,000 rpm for 5 min at 4 °C. The supernatant was discarded and the total protein obtained. Simple Western analysis was performed using the Wes Simple Western (Protein Simple) system. This assay uses an automated Western system—no gels, no transfer devices, no blots, no film and no manual analysis, resulting in a digital result. The test results were analyzed using the Compass program.

### 4.8. Apoptosis Assay

The apoptosis rate was measured with an Annexin V-FITC Apoptosis Detection Kit (Vazyme, Nanjing, China), according to the manufacturer’s instructions. After transfection, the cells were collected and washed twice with PBS, then centrifuged at 300 g, 4 °C for 5 min. The cells were resuspended in 100 μL 1× Binding Buffer and mixed gently followed by the addition of 5 μL Annexin V-FITC and 5 μL PI Staining Solution, which were allowed to incubate at room temperature for 10 min in the dark. Finally, cells were added to 400 μL 1×Binding Buffer and mixed gently. Cells were sorted by fluorescence-activated cell sorting using the Flow cytometer FACSAria SORP (Becton Dickinson, Franklin Lakes, NJ, USA). The apoptosis rate was calculated using the following equation: total number of cells is composed of number of cells in the right upper quadrant and number of cells in the right lower quadrant.

### 4.9. Determination of Melanin Level

The cells were lysed with 1 mL 0.2 mol/L NaOH. The cell lysate was collected and incubated at 37 °C for 2 h. Wavelength measurement was performed at 475 nm using a microplate reader. The standard curve was plotted using the Melanin synthetic standard (Sigma). Each group was repeated 3 times, from which the melanin level was calculated.

### 4.10. Luciferase Vector Construction and Reporter Assays

Promoter-specific primers were designed using O1igo7 (Table 4), and the *Slc7a11* promoter region was analyzed using PROMO (http://www.cbs.dtu.dk/services/Promoter/) to obtain possible transcription factor binding sites. The *Slc7a11* promoter was reconstructed into the pGL3-basic vector with restriction enzymes KpnI and BgIII. The internal reference plasmid pRL-TK and the recombinant plasmid were co-transfected into RAB-9 cells (ATCC), with the pGL3-basic plasmid and pRL-TK plasmid co-transfected cells as the negative control group and the cells with no substance transfected as the blank group. The transfected cells were collected and analyzed using the Dual-Luciferase Reporter Assay System (Promega, Madison, WI, USA).

### 4.11. Electrophoresis Mobility Shift Assay (EMSA)

The nuclear proteins of melanocytes were extracted and the concentrations determined. Based on the binding sequence of *POU2F1*, normal and mutant probes were designed (Table 5) and biotinylated at the 5′ end. The EMSA reaction system was formulated as shown in Table 6. The cold-competitive EMSA reaction system is shown in Table 7. The samples were analyzed by Native-PAGE, transferred, and UV cross-linked prior to carrying out the chemiluminescence reaction, development, and photographing.

### 4.12. Statistical Analysis

Each experiment was repeated at least three times. Statistical analysis Q-Q plot was used to check the normality of the data. Statistical significance between experimental and control groups was analyzed by Independent-Sample Test and one-way ANOVA. The results are presented as mean ± standard deviation (SD) at two levels of significance, *p* < 0.05 and *p* < 0.01.

## 5. Conclusions

In summary, rabbit melanocytes were isolated and identified. The Slc7a11 expression levels in the protein yellow colored skin tissue was higher than those from other fur colors. It was found that *Slc7a11* significantly affected the protein and mRNA expression of TYR and MITF, inhibiting melanocyte apoptosis and affecting melanogenesis by melanocytes. Furthermore, we that *POU2F1* regulated the activity of the rabbit *Slc7a11* promoter. Our results provide a theoretical basis for further exploration of the role *Slc7a11* plays in pigmentation.

## Figures and Tables

**Figure 1 ijms-20-02493-f001:**
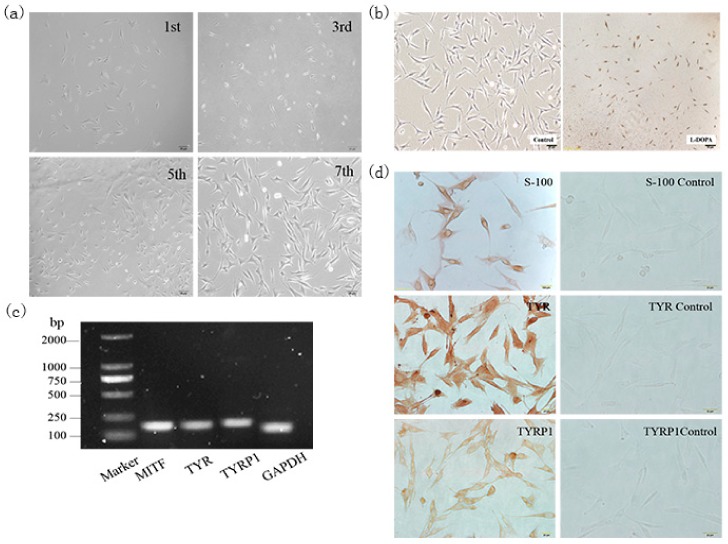
The separation and identification of melanocytes of rabbits. (**a**) Morphology of the 1st, 3rd, 5th, and 7th generation melanocytes isolated by the two-step enzyme digestion method (100×). After 12 h of isolation and culture, cobblestone-like keratinocytes and bi-polar dendritic melanocytes were observed. As the cells grew, the keratinocytes gradually metastasized while the melanocytes continued to divide, and the cell culture became more pure. (**b**) Identification of isolated rabbit melanocytes by L-DOPA staining. The 3rd generation melanocytes were treated with L-DOPA to detect the distribution of brown or black particles in isolated cells (40×). Control was treated with PBS (100×). (**c**) Real-time PCR was used to detect the expression of melanocyte-specific genes such as *MITF*, *TYR*, and *TYRP1* in isolated cells. (**d**) Isolated rabbit melanocytes were identified by immunocytochemical staining (100×) using melanocyte-specific marker proteins S-100, TYR, and TYRP1 to analyze the expression pattern of these three proteins in the isolated cells.

**Figure 2 ijms-20-02493-f002:**
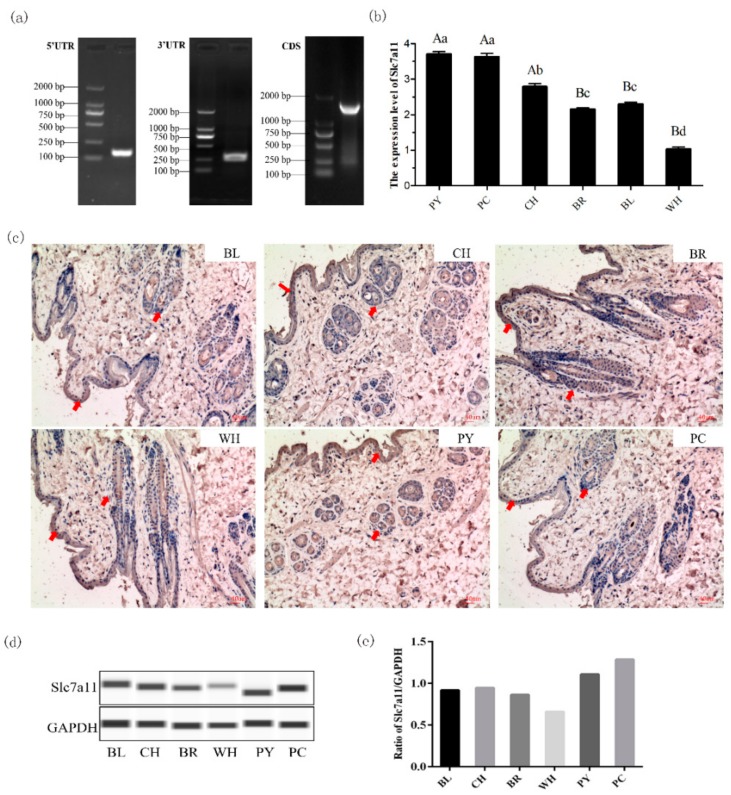
Cloning of rabbit *Slc7a11* gene and its expression in Rex Rabbit coat with different fur colors. (**a**) Full length sequence of rabbit *Slc7a11* cDNA was obtained using the RACE technique. The 5′UTR and 3′UTR sequences were obtained by 5′ RACE and 3′ RACE, respectively, and the DNASTAR program was used to assemble the sequence as well as remove redundant sequences to obtain the Rex Rabbit *Slc7a11* cDNA sequence. (**b**) mRNA expression level of *Slc7a11* gene in skin tissues of Rex Rabbits with different fur colors by Real-time PCR. Values with different capital letter superscripts signify an extremly significant difference (*p* < 0.01), and values with different small letter superscripts signify a significant difference (*p* < 0.05). (**c**) Localization of *Slc7a11* in the skin of Rex Rabbits with different fur color using immunohistochemical staining. Arrows indicate positive expression of *Slc7a11* in the epidermis and hair follicles (×100). (**d**) The expression level of *Slc7a11* (xCT) in skin tissues of Rex Rabbits with different fur colors by the Wes method. (**e**) The results were analyzed using the Compass program and the relative expression ratio of *Slc7a11* was calculated.

**Figure 3 ijms-20-02493-f003:**
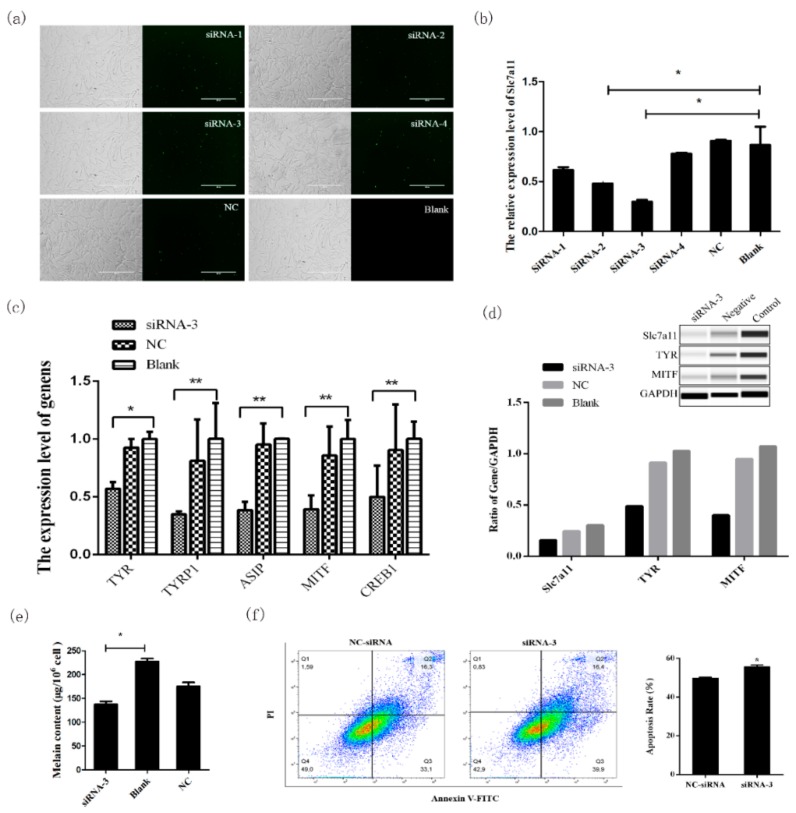
Melanogenesis-related gene expression and melanogenesis were inhibited by *Slc7a11* knockdown. An extremly significant difference was signified with “**” (*p* < 0.01), and a significant difference was signified with “*” (*p* < 0.05). (**a**) Cell morphology 24 h after transfection of melanocytes by FAM-siRNA. Melanocytes in the logarithmic growth phase were transfected and the transfection was detected by the observation of green fluorescence. (**b**) Real-time PCR detection of *Slc7a11* mRNA expression after siRNA interference. The best siRNA was screened for subsequent experiments. (**c**) Effects of *Slc7a11* interference on the expression of pigmentation-related genes such as *MITF*, *TYR*, *TYRP1*, *CREB1*, and *ASIP*. (**d**) Detection of the expressions of MITF, TYR, and Slc7a11 (xCT) proteins in melanocytes by Wes. The relative expression levels of MITF, TYR, and Slc7a11 (xCT) proteins were calculated and analyzed by the Compass program. (**e**) The effect of *Slc7a11* knockdown on melanogenesis in melanocytes. Melanocytes were collected after siRNA-3 transfection and the melanin level was measured using a microplate reader. (**f**) Melanocyte apoptosis rate was determined after knockdown of *Slc7a11*.

**Figure 4 ijms-20-02493-f004:**
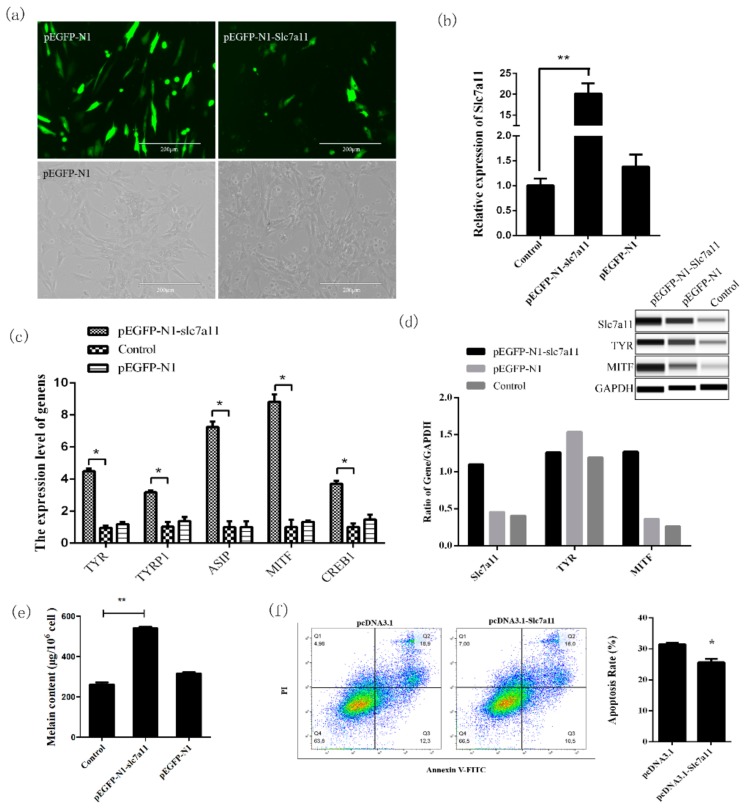
The expression of melanogenesis-related genes and melanogenesis were increased by the overexpression of *Slc7a11*. An extremly significant difference was signified with “**” (*p* < 0.01), and a significant difference was signified with “*” (*p* < 0.05). (**a**) Fluorescence detection results of pEGFP-N1-Slc7a11 transfected melanocytes. pEGFP-N1 was used as a control. (**b**) Detection of the overexpression of *Slc7a11* in melanocytes by Real-time PCR. (**c**) The effect of *Slc7a11* overexpression on the expression of pigmentation-related genes such as *MITF*, *TYR*, *TYRP1*, *CREB1*, and *ASIP*. (**d**) Detection of the expressions of MITF, TYR, and Slc7a11 (xCT) proteins in melanocytes by Wes. The relative expression levels of MITF, TYR, and Slc7a11 (xCT) proteins were calculated and analyzed using the Compass program. (**e**) The effect of *Slc7a11* overexpression on melanogenesis in melanocytes. Melanocytes were collected after siRNA-3 transfection and the melanin level was measured using a microplate reader. (**f**) Melanocyte apoptosis rate was determined after overexpression of *Slc7a11*.

**Figure 5 ijms-20-02493-f005:**
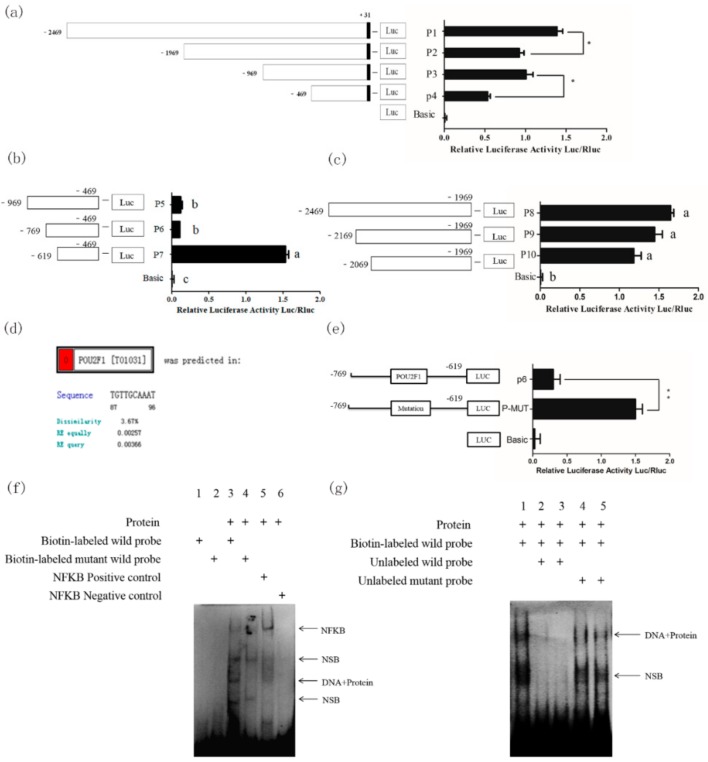
Regulation of transcriptional factor POU2F1 on *Slc7a11* promoter activity. An extremly significant difference was signified with “**” (*p* < 0.01), and a significant difference was signified with “*” (*p* < 0.05). (**a**) Preliminary analysis of the activity of *Slc7a11* promoter-deleted vector series. P1~P4 were constructed and the activity of each fragment was detected using dual luciferase. It was presumed that the *Slc7a11* promoter contained two active regions, namely −969~−469 bp and −2469~−1969 bp. (**b**) Activity detection of a series of vectors with deletions in the −969~−469 bp active region of *Slc7a11* promoter. P5–P7 were designed and constructed for this region. (**c**) Activity detection of a series of vectors with deletions in the −2469~−1969 bp active region of *Slc7a11* promoter. P8–P10 were designed and constructed for this region. (**d**) Prediction of the transcriptional binding site in the *Slc7a11* promoter region. The −769~−619bp region was the primary target based on the results of (**a**–**c**). The results also suggested that transcriptional repressors may be present in this region. The potential transcription factor binding sites were analyzed using the online program PROMO. (**e**) Site-directed mutagenesis analysis of *POU2F1*. Based on the predicted position given by PROMO, the *POU2F1* binding site was effectively mutated by site-directed mutagenesis and detected by dual-luciferase assay. (**f**) EMSA suggests that *POU2F1* binds to the *Slc7a11* core promoter region. The 1st and 2nd lanes were normal and blank mutant probes, respectively, and no bands indicated good probes. The 3rd and 4th lanes were biotin-labeled normal and mutation probes, respectively. The 5th and 6th lanes were NF-KB positive and negative controls, respectively. NSB stands for non-specific binding. (**g**) Specific binding of *POU2F1* to the *Slc7a11* core promoter region using competitive EMSA experiments. In the 2nd and 4th lanes, unlabeled probes were added at a 40:1 ratio to the labeled probes, and in 3rd and 5th lanes unlabeled probes were added at an 80:1 ratio to the labeled probes.

**Table 1 ijms-20-02493-t001:** Primers used for *Slc7a11* RACE and cloning.

Name	Sequence (5′ to 3′)	Experiment
GSP1	CACTTTCTCCTGCCCA	5′RACE
GSP2	GGCTCCTTGCCACCCAT	5′RACE
GSP3	TCCCGTTCACGTTTCCC	5′RACE
C048-1	CCCTTTCCCTTTATTCGGACCCAT	3′RACE
C048-2	CGGGGTCCCTGCCTACTATCTCTT	3′RACE
*Slc7a11* CDS-F	ATGGTCAGAAAACCTGTTGTGTCCACC	CDS amplification
*Slc7a11* CDS-R	TTATTTACGACAGTCTTCTTCAGGTACA	
*Slc7a11*-F	*CCC***AAGCTT**ATGGTCAGAAAACCTGTTGTGTCCACC	Vector construction
*Slc7a11*-R	*TCC***CCGCGG**TTATTTACGACAGTCTTCTTCAGGTACA	

Note: Bold letters indicate restriction enzyme cutting sites, italics indicate protective base.

**Table 2 ijms-20-02493-t002:** The primer sequences of *Slc7a11* siRNA primers.

Name	F/R	Sequence (5′ to 3′)
Negative Control	F	UUCUCCGAACGUGUCACGUTT
	R	ACGUGACACGUUCGGAGAATT
siRNA-1	F1	CCAUUAUCAUUGGCACCAUTT
	R1	AUGGUGCCAAUGAUAAUGGTT
siRNA-2	F2	GCAGCGACUGCUGUGAUAUTT
	R2	AUAUCACAGCAGUCGCUGCTT
siRNA-3	F3	GCAGUGAUGGUCCUAAAUATT
	R3	UAUUUAGGACCAUCACUGCTT
siRNA-4	F4	CCAUGAUUCAUGUCCGCAATT
	R4	UUGCGGACAUGAAUCAUGGTT

**Table 3 ijms-20-02493-t003:** Primer sequences used for real-time PCR.

Name	Sequence (5′ to 3′)	Product Length/bp	Tm/°C
*Slc7a11*-RT-F	TCACCATTGGCTACGTGCT	142	60
*Slc7a11*-RT-R	GCCACAAAGATCGGAACTGCT		
*GAPDH*-RT-F	CACCAGGGCTGCTTTTAACTCT	155	60
*GAPDH*-RT-R	CTTCCCGTTCTCAGCCTTGACC		
*CREB1*-RT-F	CCTCCCCAGCACTTCCTACACA	158	58
*CREB1*-RT-R	TTCAGCTCCTCAATCAGCGTCT		
*TYR*-RT-F	ATTTTCCTCGAGCCTGTACCTCC	147	59
*TYR*-RT-R	GCCAAGACTCCCGTTCATCCAC		
*TYRP1*-RT-F	CCGTCTTCTCTCAATGGCGAGT	154	61
*TYRP1*-RT-R	TGCACCATCGGTCTAGCCACA		
*ASIP*-RT-F	CTGTGCTTCCTCACTGCCTATAGCC	187	60
*ASIP*-RT-R	TTCAGCGCCACAATGGAGACCGAA		
*MITF*-RT-F	AGCTTGCCATGTCCAAACCAG	165	58
*MITF*-RT-R	TTCATACTTGGGCACTCGCTCT		

**Table 4 ijms-20-02493-t004:** Primers used for construction of *Slc7a11* promoter deletion plasmids.

Name	Sequence (5′ to 3′)	Product Length/bp	Usage
P1-F	*GG***GGTACC**ACTTTGGGATTGGTGCAG	500	P1
P1-R	*GA***AGATCT**AGTGATACCAGGGCAAAA		
P2-F	*GG***GGTACC**TTCTTCCACTTAGTCCAG	1000	P2
P2-R	*GA***AGATCT**GTAGTGATACCAGGGCAA		
P3-F	*GG***GGTACC**TGAAAGTCTGAATTAAACACA	2000	P3
P3-R	*GA***AGATCT**TACCAGGGCAAAAAGACAACA		
P4-F	*GG***GGTACC**CCTCCCAAGGTACACAGT	2500	P4
P4-R	*GA***AGATCT**AGGGCAAAAAGACAACAC		
P5-F	*GG***GGTACC**AATTTCTTCCACTTAGTCCA	500	P5
P5-R	*GA***AGATCT**TCAGAACAGACCTTCAGAGA		
P6-F	*GG***GGTACC**GATTTAACTGTGTTGCTAGG	300	P6
P6-R	*GA***AGATCT**TCCCAAAGTAGTAAATTCAG		
P7-F	*GG***GGTACC**ATATTTTTACTTGTGAGTTTAGGAA	150	P7
P7-R	*GA***AGATCT**TCAGAACAGACCTTCAGAGAGATTG		
P8-F	*GG***GGTACC**TTGCTGCCTCCCAAGGTACAC	500	P8
P8-R	*GA***AGATCT**CCCTCTCTGCCTTTCTCTCTG		
P9-F	*GG***GGTACC**CTTTCCACCCACACCTACCCT	200	P9
P9-R	*GA***AGATCT**CCTCTCTGCCTTTCTCTCTGT		
P10-R	*GG***GGTACC**ATATTTAATGGTGTGTGTAA	100	P10
P10-R	*GA***AGATCT**GTGTTTAATTCAGACTTTCA		

Note: Bold letters indicate restriction enzyme cutting sites, italics indicate protective base.

**Table 5 ijms-20-02493-t005:** Normal and mutant probes for *POU2F1*.

Name	Sequence (5′-3′)
Normal probes	TGCTTGTTGCAAATAGTCTAGCTAGACTATTTGCAACAAGCA
Mutant probes	TGCTGTCGCAGATAGTCTAGCTAGACTATCTGCGACAGCA

Note: Bold letters indicate mutation sites.

**Table 6 ijms-20-02493-t006:** EMSA reaction system.

Ingredient	1	2	3	4	5	6
PolydI:dC/μL	0.5	0.5	0.5	0.5	0.5	0.5
10× Binding Buffer/μL	0.0	0.0	1.5	1.5	1.5	1.5
Nuclear protein/μL	0.0	0.0	2.0	2.0	2.0	2.0
Biotin-labeled normal probe/μL	0.5	0.0	0.5	0.0	0.0	0.0
Biotin-labeled mutant probe/μL	0.0	0.5	0.0	0.5	0.0	0.0
Positive nuclear extract NF-KB	0.0	0.0	0.0	0.0	0.5	0.0
Negative nuclear extract NF-KB	0.0	0.0	0.0	0.0	0.0	0.5
ddH_2_O/μL	14.5	14.5	11.5	11.5	11	11

**Table 7 ijms-20-02493-t007:** Cold-competitive EMSA reaction system.

Ingredient	1	2	3	4	5
Unlabeled probe/labeled probe		40:1	80:1	40:1	80:1
10× Binding Buffer/μL	1.5	1.5	1.5	1.5	1.5
Nuclear protein/μL	2.0	2.0	2.0	2.0	2.0
Biotin-labeled normal probe/μL	0.5	0.5	0.5	0.5	0.5
Unlabeled normal probe	0.0	1.0	3.0	0.0	0.0
Unlabeled mutant probe	0.0	0.0	0.0	1.0	3.0
ddH_2_O/μL	11.0	10.0	8.0	10.0	8.0

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
