# Peer review of "Slc7a11 Modulated by POU2F1 is Involved in Pigmentation in Rabbit"

_ijms, 2019, doi:10.3390/ijms20102493_

Reviewer 1 Report

Submitted for review article entitled “"Slc7a11 modulated by POU2F1 is involved in pigmentation in rabbit".” is an original paper.

The authors try to explain the function of the Slc7a11 in melanogenesis and analysis of the mechanism of fur pigmentation in rabbit. In my opinion this is interesting and important approach to the topic but for now, these are only preliminary studies. Literature reports, that the xCT transporter encoded by the Slc7a11 gene is crucial for the regulation of pigments and can directly affect the increase of pheomelanin. Abstract and Introduction is reasonably clear. The methodology is the methodology is very limited and should be extended; especially in section RT-PCR (what were the reaction conditions) or apoptosis assay.

The interpretation of the results is clearly presented and it is adequately supported by the evidence adduced but please exchange the gels in Figure 2d, figure 3d and figure 5 g because they are a little strange. The analysis of the data is systematic and simple descriptive statistics- satisfying but please explain what test has been used by the authors to check the normality-please added in statistical section. The conclusions are logically valid and justified by the evidence adduced. All the tables and figures are adequate and necessary. Discussion is short and I suggest that the results should be discussed a little more with literature.

I have a few comments which are submitted below:

My comments:- the authors should change the gels in article!

- the authors should polish up he English language and correct some tiny grammar mistakes

- the authors should discuss results more with the literature.

Author Response

Dear Editors and Reviewers,

Thank you for your letter and for the reviewers’ comments concerning our manuscript entitled “Slc7a11 modulated by POU2F1 is involved in pigmentation in rabbit” (ID: ijms-501781). Those comments are all valuable and very helpful for revising and improving our paper, as well as the important guiding significance to our researches. We have studied comments carefully and have made correction which we hope meet with approval. Revised portion are marked in red in the paper. The main corrections in the paper and the responds to the reviewer’s comments are as following.

Responds to the reviewer’s comments:

Reviewer 1

Submitted for review article entitled “"Slc7a11 modulated by POU2F1 is involved in pigmentation in rabbit".” is an original paper.

The authors try to explain the function of the Slc7a11 in melanogenesis and analysis of the mechanism of fur pigmentation in rabbit. In my opinion this is interesting and important approach to the topic but for now, these are only preliminary studies. Literature reports, that the xCT transporter encoded by the Slc7a11 gene is crucial for the regulation of pigments and can directly affect the increase of pheomelanin. Abstract and Introduction is reasonably clear. The methodology is the methodology is very limited and should be extended; especially in section RT-PCR (what were the reaction conditions) or apoptosis assay.

Response: Based on reviewer's comments, I have added some details of the methodology.

Real-time PCR was carried out using SYBR® Premix Ex TaqTM II (TaKaRa) on an Applied Bio systems 7500 Real-Time PCR System with the following program: 1 cycle at 95°C for 30 s, followed by 40 cycles of 95°C for 5 s, and 60°C for 30 s. The primers used for detection of gene expression are shown in Table S3. Each sample was measured three times, and the results were normalized to GAPDH.

The apoptosis rate was measured with an Annexin V-FITC Apoptosis Detection Kit (Vazyme, China), according to the manufacturer’s instructions. After transfection, the cells were collected and washed twice with PBS, then centrifuged at 300g, 4 for 5 min. The cells were resuspended in 100 μL 1×Binding Buffer and mixed gently followed by the addition of 5 μL Annexin V-FITC and 5 μL PI Staining Solution, which were allowed to incubate at room temperature for 10 min in the dark. Finally, cells were added to 400 μL 1×Binding Buffer and mixed gently. Cells were sorted by fluorescence-activated cell sorting using the Flow cytometer FACSAria SORP (Becton Dickinson, USA). The apoptosis rate was calculated using the following equation: total number of cells is composed of number of cells in the right upper quadrant and number of cells in the right lower quadrant.

The interpretation of the results is clearly presented and it is adequately supported by the evidence adduced but please exchange the gels in Figure 2d, figure 3d and figure 5 g because they are a little strange.

Response: We are very sorry for our unclear statement in the methodology. Western blotting in this study was performed using Wes Simple Western system. This assay was an automated Western - no gels, no transfer devices, no blots, no film and no manual analysis. The results presented were not the same as the traditional Western blotting glue map. The band obtained by Wes Simple Western system was digital and square, while those obtained by traditional Western blotting are smooth. It was supplemented in Materials and Methods..

Attached: company website and manual. Wes Simple Western URL: https://www.proteinsimple.com/

Meanwhile, I tried my best to improve the results in the manuscript. Figure 5g was the clearest of the previous results, and I have adjusted the exposure and improved the resolution for better presentation. I hope that the correction will meet with approval.

The analysis of the data is systematic and simple descriptive statistics- satisfying but please explain what test has been used by the authors to check the normality-please added in statistical section.

Response: Considering the Reviewer’s suggestion, we have added normality analysis in statistical section. Statistical analysis Q-Q plot was used to check the normality of the data.

The conclusions are logically valid and justified by the evidence adduced. All the tables and figures are adequate and necessary. Discussion is short and I suggest that the results should be discussed a little more with literature.

Response: We have made correction according to the Reviewer’s comments.

Skin melanocytes are located on the basement membrane between the epidermis and the dermis. Skin melanocytes have been isolated from humans, pigs, and other animals using 0.25% trypsin and successfully cultured [14, 15]. In a preliminary study, the back skin of rabbits was digested with trypsin, but the results were not satisfactory, indicating that the use of trypsin alone did not result in sufficient digestion or separation of melanocytes from the skin of hairy animals. However, the use of Dispase II enzyme digestion, allowed us to obtain pure melanocytes for the first time, which laid the foundation for our later work.

TYR is a key enzyme in melanin formation and its expression level and activity determine the rate and yield of melanin production. Upon activation, TYR catalyzes the hydroxylation of tyrosine to L-3,4-phenylalanine (DOPA), which is rapidly oxidized to form dopaquinone [27]. MITF regulates the expression of the tyrosine gene family and participates in melanin production [28]. Studies have shown that the expression of TYRP1, ASIP and CREB1 determines skin melanin deposition [29-31]. The Slc7a11 gene affects the transcription of genes involved in melanogenesis, which is closely related to the formation of fur color. Moreover, Slc7a11 decreases melanocyte apoptosis and further affects melanogenesis of melanocytes. These results confirmed that Slc7a11 is closely related to the formation of Rex rabbit fur color. The regulatory factors required for such expression patterns would be the next research objective.

I have a few comments which are submitted below:

My comments:- the authors should change the gels in article!

Response: We tried our best to improve the results in the manuscript. We appreciate for your warm work earnestly, and hope that the correction will meet with approval.

-the authors should polish up the English language and correct some tiny grammar mistakes

Response: Considering the Reviewer’s suggestion, we send the revised MS to a professional English editing company, Mogo Internet Technology Co., LTD. to address the language problem. This manuscript has been proofread and edited by a native speaker.

- the authors should discuss results more with the literature.

Response: We have made correction according to the Reviewer’s comments.

Once again, thank you very much for your comments and suggestions.

Best regards

Xinsheng Wu, PhD

Professor and Dean

College of Animal Science and Technology, Yangzhou University

48 East Wenhui Road, Yangzhou 225009, PR China

Reviewer 2 Report

This is an interesting paper, which would require minor revisions, see below.

In the MM state that the studies were approved by the Local Ethics Committee. It is said that it was done according the standards but information that it was approved should be included.

Higher magnification of histological pictures will be appreciated by the readers.

The readers would also appreciate an information on the role of L-DOPA and L-tyrosine in the regulation of melanogenesis (Pigment Cell Melanoma Res 25, 14-27, 2012)

Author Response

Dear Editors and Reviewers,

Thank you for your letter and for the reviewers’ comments concerning our manuscript entitled “Slc7a11 modulated by POU2F1 is involved in pigmentation in rabbit” (ID: ijms-501781). Those comments are all valuable and very helpful for revising and improving our paper, as well as the important guiding significance to our researches. We have studied comments carefully and have made correction which we hope meet with approval. Revised portion are marked in red in the paper. The main corrections in the paper and the responds to the reviewer’s comments are as following.

Responds to the reviewer’s comments:

This is an interesting paper, which would require minor revisions, see below.

In the MM state that the studies were approved by the Local Ethics Committee. It is said that it was done according the standards but information that it was approved should be included.

Response: We have made correction according to the Reviewer’s comments in Materials and Methods.

After tissue collection, the wounds were treated with iodine tincture. After the experiment, rabbits were anesthetized by an intraperitoneal injection of sodium pentobarbital (50 mg/kg).

Higher magnification of histological pictures will be appreciated by the readers.

Response: Considering the Reviewer’s suggestion, we have justified histological pictures.

The readers would also appreciate an information on the role of L-DOPA and L-tyrosine in the regulation of melanogenesis (Pigment Cell Melanoma Res 25, 14-27, 2012)

Response: I am very grateful to the reviewers’s good comments. This reference is very valuable and has been cited in Materials and Methods.

Once again, thank you very much for your comments and suggestions.

Best regards

Xinsheng Wu, PhD

Professor and Dean

College of Animal Science and Technology, Yangzhou University

48 East Wenhui Road, Yangzhou 225009, PR China

Round  2

Reviewer 1 Report

The authors corrected manuscript according to the reviewer suggestions.